# Changes in Body Composition and Body Image Perception in Adolescent Soccer Players Examined with Repeated Measurements During Pre-Season and In-Season Training

**DOI:** 10.3390/jfmk10020219

**Published:** 2025-06-07

**Authors:** Luciana Zaccagni, Mattia Reggiani, Stefania Toselli, Emanuela Gualdi-Russo

**Affiliations:** 1Department of Neuroscience and Rehabilitation, Faculty of Medicine, Pharmacy and Prevention, University of Ferrara, 44121 Ferrara, Italy; luciana.zaccagni@unife.it (L.Z.); mattia.reggiani@edu.unife.it (M.R.); emanuela.gualdi@unife.it (E.G.-R.); 2Center for Exercise Science and Sports, University of Ferrara, 44123 Ferrara, Italy; 3Department for Life Quality Studies, University of Bologna, 47921 Rimini, Italy

**Keywords:** adolescence, football, body dissatisfaction, body fat, body fat-free mass

## Abstract

**Objectives**: Adolescents’ health is positively influenced by the performance of physical activity. Regarding soccer, a very popular sport, the aims of the study were to assess changes in body composition and body image of late adolescent players during pre- and in-season training periods, analyzing the relationships between dissatisfaction and body composition parameters. **Methods**: A sample of 16–19-year-old male soccer players was examined longitudinally by three surveys. The body composition was assessed by anthropometric method. Body image perception was evaluated using two different figural scales related to shape and muscularity. **Results**: During the period examined, fat-free mass increased, and changes in perceived and ideal figures occurred, showing a desire toward more robust and muscular figures. Body image dissatisfaction was generally low, with a decrease in muscle dissatisfaction during the period. Body composition parameters significantly predicted body image dissatisfaction. **Conclusions**: Our findings suggest anthropometric and body image perception changes during soccer training with improvements in body composition parameters and a decrease in musculature dissatisfaction. These results highlight the importance of incorporating physical and psychological monitoring into training programs to support the healthy development of athletes’ body image and body composition.

## 1. Introduction

Physical activity (PA) is deemed to positively affect physical and mental health [1]. This is especially important during adolescence, a critical period for developing a positive or negative body image perception and ideals, as it is characterized by puberty and rapid and diverse physical changes in body shape, size, composition, and identity development.

Body image (BI) refers to people’s subjective view of their bodies, including their thoughts, perceptions, and feelings. When the person does not achieve their ideals, body image dissatisfaction (BID) emerges. In childhood, BID is associated with negative physical and mental health outcomes that include a rise in depressive symptoms, poor self-esteem, obesity and overweight, eating disorders, physical inactivity, and poor fitness [2]. During adolescence, achieving a positive self-concept and fostering mental wellbeing can be facilitated by engaging in PA, which improves body perceptions and satisfaction [3]. On the contrary, BID, teasing, and problems with gender identification may become determinants for continuing PA, more so than actual motor skills [4]. It is well known that a physically active lifestyle during developmental age has favorable repercussions in adulthood [5] and improves cardiovascular health, self-confidence, muscle strength and endurance, bone density, and body weight [6]. In general, increased participation in PA and sports promotes a positive BI: BID decreases as the number of hours spent in sports activities increases [7] and increases with increasing BMI and body fat accumulation [8]. Physical inactivity and sedentary behavior are believed to be the main drivers of the increasing prevalence of overweight/obesity in Europe [9]. Higher PA levels were found to be associated with lower BMI values, better cardiorespiratory fitness, and lower body dissatisfaction [10].

Adolescent participation in team and individual sports is associated with better physical, mental, social, and emotional health [11]. In particular, adolescents participating in team sports appear to have better psychosocial health (lower anxiety, depression, and social problems) than those participating in individual sports [12].

Soccer is one of the most popular team sports, involving players of all ages at amateur and professional levels [13]. Therefore, it is important to understand possible physical and mental health risks for soccer players with a focus on youth groups. BID can lead to several negative health outcomes and the potential establishment of psychopathologies [14]. Another relevant element to evaluate in the sportsman is to consider not only body shape but also muscle development. In this specific case, a scale related to muscle development has been proposed for adolescents in addition to the one related to form [15]: the new figure-rating scales allow rapid and robust assessment of both aspects. In addition to the specific case of adolescents engaged in sports, the use of these differentiated scales is appropriate because eating disorders would arise as a response to dissatisfaction related to body fat and not muscularity [16]. In general, using the two scales makes it possible to distinguish whether most of the BID concerns the thin, lean ideal. While in the female sex there is generally a preference toward an ideal of thinness (‘drive for thinness’), in the male sex there may be a preference for a lean and toned body (‘drive for leanness’) or a larger and more muscular body [15].

Despite the importance of these aspects for adolescents’ good health, there is limited research on BI and body composition or weight status in adolescents [7,17], and, in particular, research is lacking regarding trends in body composition parameters and BID in soccer players. Although we could verify that pre-adolescent soccer players showed anthropometric and body image perception changes after a 12-week training program [18], with a positive effect of sports on BI, an analogous longitudinal study during late adolescence has not been conducted to date. The main aims of the present study were as follows: i. To examine changes in body composition and BI perception throughout the pre-season and in-season soccer training in adolescent soccer players; ii. To test for any relationship between BID and body composition parameters.

## 2. Materials and Methods

### 2.1. Participants and Procedure

This longitudinal study was carried out after the approval by the Bioethics Committee of the University of Bologna (approval code: 25027; 13 March 2017) on a sample of 28 soccer players corresponding to the minimum sample size expected through an a priori power analysis using the G*Power statistical program (version 3.1.9.6; Universitat Kiel, Kiel, Germany) for 80% power, medium effect size, and a 0.05 significance level for performing a primary analysis (repeated measures ANOVA) according to Andrade [19]. Participation in the research was bound by informed consent, which was signed directly by participants over 18 years of age and by parents in the case of minors.

The non-elite group consisted of players aged 16–19 who participated in the under-18 teams in the Emilia-Romagna second-tier soccer league in 2023–2024. All thirty boys aged ≥16 registered with a soccer club, selected on a convenience basis, were invited to participate in the three surveys conducted every three months (first: September 2023; second: December 2023; third: March 2024). Thirty players voluntarily agreed to participate, but two of them could not take part in all the surveys because they were injured. The final sample was therefore 28 players, and there were no missing values in the data collected during the surveys of this final sample. The average age of beginning organized soccer activities was 7.4 ± 3.3 years in the surveyed sample. All players trained for five hours a week (subdivided into three training days), plus a match at the end of the week during the soccer in-season period.

### 2.2. Anthropometric Traits

A trained operator performed all anthropometric surveys following standard procedures [20,21]. Anthropometric measurements were taken on players in light clothing and without shoes. The anthropometric traits measured were the following: stature, weight, and skinfold thicknesses (triceps, subscapular). Stature was measured to the nearest 0.1 cm using an anthropometer (Magnimeter, Raven Equipment Ltd., Dunmow, Essex, UK) on participants with their heads aligned with the Frankfurt plane in the standing position. Weight was measured to the nearest 0.1 kg using a digital scale (SECA, Basel, Switzerland). Skinfold thicknesses were taken at the triceps and subscapular points to the nearest 0.5 mm on the left side of the body (according to Weiner and Lourie [22]) by a Lange caliper (Beta Technology Inc., Houston, TX, USA). The operator’s TEMs (assessed before the survey) were <5% for skinfolds and <1% for other measurements.

Some body composition parameters were calculated using the anthropometric measurements that were taken. Body Mass Index (BMI) was computed as weight (kg)/height^2^ (m^2^). Using BMI, we classified participants according to Cole’s cut-offs [23,24] into four weight status categories: underweight, normal weight, overweight, and obese. The sum of triceps and subscapular thickness skinfolds was used to obtain body density by Durnin and Womersley’s equation [25], and then the percentage of body fat (%Fat) by Siri’s equation [26]. Subsequently, fat mass in kg (FM) and fat-free mass (FFM) in kg were obtained.

### 2.3. Body Image Perception and Dissatisfaction

The Male Body Scale (MBS) and Male Fit Body Scale (MFBS), developed and validated by Ralph-Nearman and Filik [15], were submitted to the participants to assess for each scale the perceived current body figure (actual) and ideal body figure (ideal) related to body fat (MBS) and muscularity (MFBS). MBS includes nine male figures that vary progressively from emaciated to obese; MFBS, in turn, includes nine male figures ranging from very lean to very muscular. Using the first scale to derive the current and ideal figures, it was possible to derive the adipose-related BID, while the second one allowed for indications of muscle-related BID.

Participants were successively presented with the two scales, with the figures numbered from 1 to 9, and had to indicate the number corresponding to the chosen figures in response to the following questions asked first on the MBS and then on the MFBS: 1—Which figure do you think best represents you? 2—Which figure would you like to resemble? 3—Which figure do you think corresponds to the ideal soccer player?

Answers to the first question were reported as perceived shape (S-Feel) and perceived muscularity (M-Feel), and those to the second question as ideal shape (S-Ideal) and ideal muscularity (M-Ideal). The answers to the third question were given as ideal soccer player shape (S-Soccer) and ideal soccer player muscularity (M-Soccer).

The degree of BID was determined through the discrepancy between the actual and ideal figure (FID or Feel minus Ideal Discrepancy): the index was obtained by subtracting the ideal figure score from the actual one, thus resulting in a discrepancy relative to shape (S-FID) and a discrepancy relative to muscularity (M-FID) according to the used scale. Finally, the discrepancy from the ideal soccer player figure (FIDsport, according to [27,28]) was calculated by subtracting the score of the ideal figure of the soccer player from the actual figure of the boy examined on both the shape and muscularity scales (S-FIDsport and M-FIDsport). Generally, the FID score will be positive when the actual figure is larger than the ideal one and negative when the actual figure is thinner/leaner than the ideal one. If there is no discrepancy (the figure chosen as the real one matches the ideal one), the FID score will be 0.

### 2.4. Statistical Analysis

Assumptions of normality were verified using Kolmogorov–Smirnov tests. Comparisons of skinfold thicknesses among surveys were performed after their log transformation.

Mean and standard deviation (SD) were used to describe continuous variables, and percentage frequencies to describe categorical variables.

Longitudinal anthropometric and BI perception changes were assessed by analysis of data collected in the three successive surveys (repeated measures ANOVA). A nonparametric Friedman rank-sum test was applied to three successive surveys for variables related to BI perception (not normally distributed). The effect size was calculated using partial eta–squared for repeated measures analysis of variance and using Kendall’s W coefficient of concordance for the Friedman test.

Linear multiple regression analyses were conducted between the BID (dependent variable) and the independent variables of body composition (BMI, %Fat, FFM). The analyses were preceded by a check for multicollinearity using the variance inflation factor (VIF).

All statistical analyses were computed using STATISTICA software, version 11 (StatSoft, Tulsa, OK, USA).

## 3. Results

Table 1 shows the mean values and SD of anthropometric traits of the soccer player sample and the statistical comparisons among the three surveys.

Significant changes were observed for stature, weight, subscapular skinfold thickness, and FFM with increasing values during the period under consideration and a large effect size (partial eta-squared > 0.14). Some of these changes can be attributed to the normal growth process (e.g., statural increase); others are attributable to both the growth process and the effect of sports training (e.g., weight, FFM). Unlike FFM (+1.2 kg), the FM and %Fat showed non-significant changes during the soccer pre-season and in-season.

Considering the weight status, no differences in frequency were found over the period considered. The most represented category was normal weight (67.9%), followed by overweight (28.6%) and underweight (3.6%). There were no obese boys in the soccer player sample.

Regarding the perception of BI, the figures most frequently selected were No. 4 as actual figures for both body shape and muscularity and as ideal figures for both body shape and soccer player shape, and No. 5 as ideal figures for both personal and soccer player muscularity. Table 2 shows the changes in BI perception and dissatisfaction during the pre-season and in-season.

BI perception variables changed significantly for the ideal body shape (S-Ideal), showing a desire for a more robust shape, and for current musculature (M-Actual), where there was a significant increase in perceived muscular figure. The figures of the ideal soccer player changed over time in both shape (S-Soccer) and musculature (M-Soccer): the former increased, the latter decreased.

Regarding BID indices, the adolescent soccer players examined generally showed low levels of dissatisfaction, with non-significant changes over the period. The only exception was the changes in FID related to the muscularity of the ideal soccer player (M-FIDsport). In general, M-FIDsport mean values were negative, indicating a desire to be more muscular to come closer to the ideal image of the soccer player. This index decreased significantly throughout repetitions in probable relation to the player’s awareness of increased musculature with soccer training, consistent with the concomitant significant increase in M-Actual.

Table 3 shows the results obtained by the multiple regression analyses: the three independent variables analyzed were significant predictors of FID, indicating an increase in dissatisfaction as BMI and %Fat increased and a decrease in FID as FFM increased. The obtained models explain from more than 30 to more than 60% of the variance, emphasizing the importance of body composition parameters for BI purposes. In the S-FID, only BMI was a significant predictor in the third survey, implying an increase in BID as BMI increased.

Overall, the models constructed from the second and third surveys are highly significant, indicating that body composition parameters explain 48% and 61% of the variance in shape dissatisfaction, respectively. The models obtained by examining M-FID capture a significant effect only at the third survey: as BMI increases and %Fat decreases, dissatisfaction about muscularity increases. This model explains 34% of the variance. Dissatisfaction with the ideal soccer player shape has BMI as a significant predictor (in the second and third surveys): dissatisfaction increases as BMI increases. The three models obtained based on body composition were highly significant, going on to explain 48%, 40%, and 52% of the variance, respectively. Finally, dissatisfaction concerning muscle development of the ideal soccer player led in the third survey to a highly significant model based on the three predictors considered (BMI, %Fat, FFM): dissatisfaction increases as BMI increases and %Fat and FFM decrease. This model explains more than 48% of the variance.

## 4. Discussion

The current study aimed to detect any changes in body composition and BID perception of adolescent soccer players examined longitudinally, analyzing the associations between BID and body composition parameters.

The main findings showed significant changes in both anthropometric traits and BI perception in adolescent players over the year during the pre-season and in-season soccer training. Anthropometric changes are consistent with the process of growth and development, involving the completion of individual anthropometric characteristics with significant increases in stature and weight. In particular, as is well known, the statural growth, after the peak involving an average increase of 7 cm at puberty in the male sex reached on average at 14 years of age, goes down in speed, progressively decreasing from the year in which the spurt occurred and in the final stages of adolescence mainly implying an increase in the length of the trunk in comparison to the lower limb [29]. During this period, we observed the expected change in body composition with an increase in muscle mass in the male sex and a decrease in %Fat [29,30]. Indeed, although no significant changes were observed in FM and %Fat, soccer training of Italian adolescent players resulted in a significant increase in FFM, similar to what was found in Portuguese adolescents aged 15-16 years, for whom significant decreases in FM were also detected [31]. These anthropometric trends, partly related to the growth process, are connected to sports training and are consistent, in particular, with the expected effects on body composition due to soccer training [32]. Thus, a previous study comparing Italian adolescents practicing basketball versus those practicing soccer showed a lower %Fat and lower endomorphy in the latter [33]. A significant association between soccer performance and low-fat levels (particularly low in elite players) was also observed [34]. Soccer players have a slender body build compared to non-sporty individuals [33,35]. A recent systematic review and meta-analysis [32] highlighted that soccer improves children’s body composition (FM and FFM).

More generally, the results obtained are satisfactory to the extent that the inclusion of soccer among health-enhancing physical activities is appropriate, as suggested by Hernandez-Martin et al. [32].

Considering anthropometric changes, the perception of BI in a body that is changing in shape and composition may lead to changes and eventual dissatisfaction. Adolescent players examined through the dual scale proposed by Ralph-Nearman et al. [15] showed a tendency to prefer slightly more robust shapes throughout the soccer season and a more muscular body ideal. Some apparent inconsistencies, such as increasing dissatisfaction as %Fat decreases, confirm the well-known male tendency to be less aware of their weight status than the female sex and to mistakenly believe that a more robust physique corresponds to greater muscularity [36,37]. Moreover, consistent with a study of 9- to 10-year-old Italian children practicing soccer [18], this research showed a positive effect of sports on the perception of BI: adolescent players were generally satisfied with their body shape (S-FID). This trend confirms the tendency of adolescents to improve BI with sport participation [37,38], with a more positive BI in athletes than in non-athletes [39]. The effect of PA reverberates on the perception of BI and, more generally, on wellbeing, so much so that PA among young people is suggested “to improve wellbeing and yield potential health benefits” [40]. Confirming this, previous studies have found that adolescents from various nationalities have higher levels of life satisfaction related to physical activities, such as team or non-team sports, probably because of the biological, psychological, and social benefits of physical activities on wellbeing [12,41,42].

Adolescence and the transition from adolescence to adulthood represent delicate periods for BID [43]. Several previous studies contributed to showing that BID can predict the tendency toward eating disorders, with possible public health implications (among others: [44,45,46]). PA generally results in decreased body dissatisfaction at all ages [37], and, in particular, athletes of both sexes are believed to have a lower level of dissatisfaction than non-athletes [39]. Distinguishing between non-esthetic/non-lean (e.g., ball sports) and esthetic/lean sports (e.g., gymnastics), a recent review showed that lean athletes have greater concerns about body image than non-lean athletes, although the meta-analysis found no significant differences between the athlete groups [39]. However, in addition to the background of sports training, individual/non-individual competition, and training intensity, other factors could intervene to influence BI in sports, such as pressures from other people (coaches, judges, parents, and peers) and training regimens [47]. Conversely, according to Webb et al. [43], athletes’ BID with their body composition would be similar to that of the general population: their BI will be affected both by the ideal athlete practicing that sport and by the general social ideals. However, in our view, it is important to distinguish dissatisfaction with these different ideals, as the sporting ideal can be very different from the social ideal. Thus, for example, gymnasts were found to be satisfied with their BI in the social context even though they manifested dissatisfaction concerning their ideal of “gymnast” [27]. Young athletes face sport-specific pressures related to body shape and weight, increasing the risk of body dissatisfaction. These pressures include idealized body standards, critical comments, objectification, and sports regulations emphasizing physical appearance, particularly in weight-sensitive and esthetic sports (e.g., gymnastics, figure skating, long-distance running, and triathlons) [48,49,50]. Research on body image concerns has primarily focused on females, with studies on male athletes showing inconclusive results [39,51,52]. While lean athletes tend to report higher body image concerns, this pattern is observed in females, not males. Factors influencing body satisfaction in young athletes include motivation type: intrinsic and autonomous motivation are linked to better body satisfaction, while extrinsic motivation is associated with higher dissatisfaction [53,54,55,56,57]. Body satisfaction fluctuates during adolescence, with males showing an increase in satisfaction from ages 12 to 20, while females experience a decline between ages 10 and 16, which stabilizes and improves by age 20 [58]. Adolescent boys, unlike girls who typically pursue thinness, often have a desire for muscularity [36,59]. While this drive can encourage healthy habits, it can also lead to extreme, unhealthy behaviors when taken too far. Grieve [60] found that internalizing muscular ideals is strong in early adolescence. Around 25% of middle-school boys lift weights to increase muscle mass [61]. While sport can promote positive behaviors, certain sports may encourage unhealthy practices, such as weight-focused sports (e.g., wrestling, cross-country running) linked to eating disorders [62] or muscle-focused sports (e.g., football, weightlifting) associated with steroid use [63]. Symptoms of muscle dysmorphia often begin in adolescence [64].

In the sample of soccer players examined using longitudinal analysis, in the pre-season, we were able to verify a complete satisfaction with the expected body shape in the social context (S-FID), while dissatisfaction was greater than in the ideal soccer player, especially for musculature (M-FID), which has been declining, however, as described, during the sports season. The aspiration toward greater muscle mass and body fat has also been reported in a previous cross-sectional study of 19-year-old soccer players [65]. In general, high self-confidence, self-esteem, and positive BI levels of athletes would depend on enhanced self-efficacy after completing physical tasks related to the sport played and better body consciousness [12]. The representation of body image across different eras reflects shifting ideals of beauty, strength, and human potential, shaped by cultural, social, and historical influences. In sports, the human body has symbolized the values of each time period. These ideals are dynamic, evolving with societal norms and cultural beliefs, and understanding them requires considering the broader historical and cultural context [66].

The main strengths of the study consist of its longitudinal design and the anthropometric measurements taken directly by a trained operator. This study also has several limitations. The absence of a control sample of non-athletes prevented us from testing for changes attributable with certainty to the normal growth process. Although the sample size analyzed meets the minimum required by the power analysis for a longitudinal study, it did not allow us to verify any differences in BI perception concerning the role of play. Differences in the competitive level and type of sport played are relevant aspects that could be addressed in future research. In addition, future studies should examine players of various age groups and both sexes and analyze differences with end-season and/or off-season periods. Furthermore, a more explicit discussion of the potential pedagogical implications of the findings could contribute to a deeper understanding of the psychological aspects involved in body image perception in adolescent athletes and help translate the research into practical strategies for sports educators, coaches, and practitioners.

Based on our study findings, some practical applications for coaches, trainers, and youth sport practitioners should be considered. First, regular monitoring of anthropometric traits and body image perception throughout the season can help tailor training loads and psychological support to the individual needs of adolescent players. Second, since adolescents may aspire to an unrealistic muscular ideal, practitioners should promote a healthy and functional body image, emphasizing strength and performance over appearance. Third, coaches and educators should be aware of the psychosocial impact of body image dissatisfaction and include BI education and body positivity strategies as part of youth development programs. It is important for those working with adolescent boys to recognize that body dissatisfaction can lead to harmful compensatory behaviors affecting their health and development. Promoting a positive body image, self-esteem, and early intervention is crucial. Coaches, educators, clinicians, and parents should be knowledgeable about identifying signs of muscle dysmorphia and discussing healthy ways to address body image concerns. Educating adolescents about natural body differences and the dangers of unhealthy body manipulation practices and providing healthier alternatives for weight management and fitness is key.

In conclusion, our findings confirm the importance of adolescents’ participation in sports by showing physical and mental improvement resulting from playing soccer. Because improving BI and body composition parameters (increasing muscle mass and decreasing %Fat) is a relevant factor for both sports activity performance and good health perspectives, families and schools should make efforts to encourage children and adolescents’ participation in sports activities.

## Figures and Tables

**Table 1 jfmk-10-00219-t001:** Anthropometric traits and body composition changes in adolescent soccer players during the soccer pre-season and in-season.

	First Survey		Second Survey		Third Survey		ANOVA		
Variables	Mean	SD	Mean	SD	Mean	SD	F	*p*	Partial η^2^
Stature (cm)	175.0	6.8	176.0	6.7	177.0	6.8	83.30	0.001	0.755
Weight (kg)	69.9	9.9	70.7	9.75	71.7	11.2	6.64	0.003	0.197
Triceps skinfold (mm)	10.8	4.3	10.8	4.2	10.9	4.7	0.06	0.944	0.002
Subscapular skinfold (mm)	8.2	2.4	9.1	2.9	8.9	3.3	9.15	<0.001	0.253
**Indices**									
BMI (kg/m^2^)	22.8	2.9	22.8	2.8	22.9	3.2	0.42	0.662	0.015
Density (g/cc)	1.067	0.009	1.065	0.009	1.066	0.011	1.50	0.222	0.054
%Fat	14.1	4.0	14.7	4.1	14.4	4.7	1.55	0.222	0.054
Fat Mass (kg)	10.1	4.0	10.7	4.2	10.7	5.0	2.10	0.132	0.072
Fat-Free Mass (kg)	59.8	7.1	60.1	6.8	61.0	7.4	6.33	0.003	0.190

Note: Comparisons among skinfold thicknesses were performed using log skinfolds.

**Table 2 jfmk-10-00219-t002:** BI perception and dissatisfaction changes in adolescent soccer players during the soccer pre-season and in-season.

	First Survey		Second Survey		Third Survey		Friedman Test		
MBS	Mean	SD	Mean	SD	Mean	SD	F_r_	*p*	Kendall’s W
S-Actual Figure	4.1	1.1	4.1	1.0	4.3	1.1	2.377	0.305	0.042
S-Ideal Figure	4.1	0.8	4.3	0.7	4.4	0.6	6.465	0.039	0.115
S-Soccer	4.3	0.9	4.1	0.8	4.6	0.6	8.818	0.012	0.157
S-FID	0.00	1.02	−0.21	0.92	−0.11	0.92	1.705	0.426	0.030
S-FIDsport	−0.18	1.25	−0.04	1.07	−0.29	1.15	3.127	0.209	0.056
**MFBS**									
M-Actual Figure	4.0	1.2	3.9	1.0	4.4	1.2	7.841	0.020	0.140
M-Ideal Figure	5.4	1.4	5.1	1.1	5.5	1.0	3.844	0.146	0.069
M-Soccer	5.5	1.1	5.1	1.1	5.3	0.9	5.746	0.057	0.103
M-FID	−1.46	1.04	−1.25	0.84	−1.07	0.98	4.415	0.110	0.079
M-FIDsport	−1.57	1.29	−1.18	1.16	−0.93	1.33	6.727	0.035	0.120

Note: MBS: Male Body Scale; MFBS: Male Fit Body Scale; S: shape; M: muscular; FID: feel minus ideal discrepancy.

**Table 3 jfmk-10-00219-t003:** Body composition predictors of BID by multiple regressions during the soccer pre-season and in-season.

	First Survey			Second Survey			Third Survey		
	VIF	β	*p*	VIF	β	*p*	VIF	β	*p*
**S-FID**									
BMI (kg/m^2^)	5.77	0.700	0.138	5.368	0.160	0.622	5.991	0.720	0.022
%Fat	3.40	−0.249	0.484	3.192	0.409	0.111	3.230	−0.098	0.653
Fat-Free Mass (kg)	2.50	−0.276	0.366	2.400	0.322	0.146	2.739	0.201	0.320
R^2^		0.134			0.542			0.655	
R^2^ adjusted		0.027			0.484			0.612	
*p*		0.315			<0.001			<0.001	
**M-FID**									
BMI (kg/m^2^)	5.77	0.700	0.138	5.368	0.374	0.419	5.991	1.267	0.003
%Fat	3.40	−0.249	0.484	3.192	−0.059	0.867	3.230	−0.549	0.045
Fat-Free Mass (kg)	2.50	−0.276	0.366	2.40	−0.425	0.172	2.739	−0.348	0.190
R^2^		0.134			0.090			0.416	
R^2^ adjusted		0.027			0.024			0.343	
*p*		0.315			0.510			0.004	
**S-FIDsport**									
BMI (kg/m^2^)	5.77	0.268	0.430	5.368	0.756	0.038	5.991	0.837	0.017
%Fat	3.40	0.357	0.176	3.192	−0.061	0.820	3.230	−0.083	0.732
Fat-Free Mass (kg)	2.50	0.232	0.302	2.40	−0.035	0.881	2.739	0.026	0.908
R^2^		0.537			0.471			0.571	
R^2^ adjusted		0.479			0.405			0.517	
*p*		<0.001			0.001			<0.001	
**M-FIDsport**									
BMI (kg/m^2^)	5.77	0.876	0.067	5.368	0.546	0.219	5.991	1.741	<0.001
%Fat	3.40	−0.495	0.170	3.192	−0.179	0.596	3.230	−1.109	<0.001
Fat-Free Mass (kg)	2.50	−0.458	0.141	2.40	−0.028	0.924	2.739	−0.691	0.006
R^2^		0.136			0.163			0.542	
R^2^ adjusted		0.028			0.058			0.485	
*p*		0.311			0.226			<0.001	

Note: VIF: variance inflation factor; β: standardized coefficient; S-FID: feel minus ideal discrepancy relative to shape; M-FID: feel minus ideal discrepancy relative to muscularity; S-FIDsport: feel minus ideal soccer player figure discrepancy relative to shape; M-FIDsport: feel minus ideal soccer player figure discrepancy relative to muscularity.

## Data Availability

Data is available upon request due to ethical restrictions regarding participant privacy. Requests for the data may be sent to the corresponding authors.

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
