# Peer review of "Changes in Body Composition and Body Image Perception in Adolescent Soccer Players Examined with Repeated Measurements During Pre-Season and In-Season Training"

_jfmk, 2025, doi:10.3390/jfmk10020219_

Round 1

Reviewer 1 Report

Comments and Suggestions for Authors

Major concerns

About the Introduction section, authors should include research based on epidemiology of body composition and obesity and related. For instance, Miguel Angel Tapia Serrano research would be appropriate studies. So, more rationale about this issue should be included.

In addition, Effect size could be included to indicate the meaningfulness of the differences in the mean values of repeated measures ANOVA.

Minor concerns

ABSTRACT

Abstract should be written in past.

Authors should revise abstract; some points are necessary in different sentences.

KEYWORDS

You should include football instead of soccer.

METHODS

How many observations there are in total? How many training days were during weeks?

More information about participants is required. For instance, body mass or boy fat information.

RESULTS

Note should be included in Table 3. More information about some variables is necessary for readers.

DISCUSSION

The first paragraph is expected to summarize the aims and the new insights on this issue also. Then, authors could summarize the main results.

This is an important study based on important question in our society, concretely in physically active society. For that reason, you should include some practical applications or indications after considering these results for practitioners.

What about off-season period in soccer? Would be possible to analyze the differences between end-season period and the next early-season period?? Or, even, differences between end-season period and pre-season period???

Author Response

The authors thank Reviewer 1 (R1) for his/her comments and for taking the time to evaluate this manuscript. The responses of authors (A) are listed below.

R1: The English could be improved to more clearly express the research.

A: We had the English revised.

Major concerns

R1: About the Introduction section, authors should include research based on epidemiology of body composition and obesity and related. For instance, Miguel Angel Tapia Serrano research would be appropriate studies. So, more rationale about this issue should be included.

A: Thank you for your suggestion. The Introduction has been expanded to include epidemiological research and cite the research by M.A. Tapia Serrano.

R1: In addition, Effect size could be included to indicate the meaningfulness of the differences in the mean values of repeated measures ANOVA.

A: Effect size was included in the Materials and Methods (lines 167-169) and Results (Tables 1 and 2, and lines 180-181).

Minor concerns

ABSTRACT

R1: Abstract should be written in past.

A: We have modified the verb tense (line 15), as suggested.

R1: Authors should revise abstract; some points are necessary in different sentences.

A: The abstract (including punctuation) was revised. Thank you.

KEYWORDS

R1: You should include football instead of soccer.

A: Done.

METHODS

R1: How many observations there are in total? How many training days were during weeks?

A: On each participant, we have collected anthropometric variables and body image perception in the three surveys; moreover, during the first survey, we have collected information on sports practice. The soccer players trained three days a week. The required information was added to the manuscript.

R1: More information about participants is required. For instance, body mass or boy fat information.

A: The information about body mass, body fat, and body composition parameters is reported in subsection 2.2.

RESULTS

R1: Note should be included in Table 3. More information about some variables is necessary for readers.

A: Thank you for highlighting the absence of an explanatory note in Table 3. The note has now been added.

DISCUSSION

R1: The first paragraph is expected to summarize the aims and the new insights on this issue also. Then, authors could summarize the main results.

A: The first paragraph has been modified according to your suggestions (lines 242-244).

R1: This is an important study based on important question in our society, concretely in physically active society. For that reason, you should include some practical applications or indications after considering these results for practitioners.

A: Thanks for the suggestion. The manuscript has been integrated to include these aspects at lines 342-347 and from line 362 onwards.

R1: What about off-season period in soccer? Would be possible to analyze the differences between end-season period and the next early-season period?? Or, even, differences between end-season period and pre-season period???

A: Thank you for the suggestions, we will take them into account in the next surveys. We have added this gap among the study limitations (lines 353-354).

Reviewer 2 Report

Comments and Suggestions for Authors

The title is appropriate and reflects both the subject of study and the context of application. However, a slight modification is suggested to highlight the longitudinal approach of the study, to increase the article’s reach in academic search engines.

The summary summarizes the objectives, methodological approach and main results; in my view it should include a reference highlighting adequately the practical implications of the results obtained.

The introduction is well structured and contextualizes the relevance of the study; presenting a consistent theoretical background on body composition, body image and adolescence. With regard to the sources used, it is worth noting the inclusion of recent references, such as Gualdi-Russo et al. (2022) and Cabral et al. (2024); although some older references should be updated or supplemented (for example, Hausenblas & Downs, 2001) with recent meta-analyses or systematic reviews. In addition, it would be useful to incorporate studies that specifically address the relationship between body image and sporting ideal in football by reference to other collective and individual sports.

As far as the method is concerned, the longitudinal design seems to be appropriate for the objectives of the study. The sample, although limited is acceptable (n=28), and a justified sampling with power analysis is used (GPower*). The tools used (MFBS and MBS) are validated and suitable for evaluating body image in adolescents. However, it would be useful to add information on the training of instrument implementers, the order in which scales are presented and the strategy used to minimize certain biases such as social desirability. It would also be relevant to clarify whether the scales are validated for the Italian population or a similar socio-cultural context and if intercoding reliability tests or internal consistency analysis were applied.

The results section is well organized and the data are presented clearly through tables and repeated measures ANOVA analysis, which serves adequately for the purposes of the article. Interesting results can be seen in the evolution of body composition and image perception; however, there is a lack of graphs to facilitate comparative reading  (for example, bar graphs or scatters between % body fat and body image discrepancy). Therefore, although significant differences are observed, the narrative could be strengthened by directly connecting physical changes with perceived changes in self-image, thus facilitating understanding of the type and degree of relationship between these variables.

The discussion is well articulated, integrating the results with existing literature and recognizing the complexity of the body ideal, also in the sports context. The analysis of the difference between social and sporting ideals is particularly appreciated. Precisely because of the interest of this type of association, it is recommended to enrich this section with recent studies that address the specific aesthetic pressures in adolescent male athletes and the psychological implications of body configuration in general and, more specifically, the role of ideal musculature between sportsmen and non-sportsmen on the one hand, the evolution of these relationships and, advancing hypotheses that could allow to tackle studies where the role of possible gender differences is relevant. It would also be desirable, in my opinion, to include a critical reflection on methodological limitations such as the small sample size, the lack of a control group or the absence of complementary qualitative analyses while more explicitly supplementing the possible pedagogical implications of the study.

The conclusion adequately summarizes the main findings, highlighting the importance of sports practice in the physical and emotional well-being of adolescents in a socio-cultural context where lack of movement has already become a pandemic. However, it is too general and does not perhaps provide any new interpretative elements. It is suggested to expand it with concrete practical recommendations for trainers, educators and families, as well as proposing future lines of research, as the above-mentioned comparative analyses between genders and differences between sports disciplines or levels of competence. There could also be further reflection on the applicability of this methodology in educational or school contexts.

The bibliography is extensive, up-to-date and diverse, with a renewed contribution from the Introduction section. Of the total references used, more than 30% correspond to the last five years, which is in line with current editorial standards.

Based on the set of assessments made, I consider that the manuscript presents a relevant contribution to the field of physical education, health and sport psychology and it is recommended to accept the article with minor revisions.

Author Response

The authors thank Reviewer 2 (R2) for his/her helpful comments and suggestions, which enabled significant improvements to the manuscript. The authors are also grateful for the positive and conclusive comments of the reviewer. The responses of authors (A) to comments and suggestions are listed below.

R2: The title is appropriate and reflects both the subject of study and the context of application. However, a slight modification is suggested to highlight the longitudinal approach of the study, to increase the article’s reach in academic search engines.

A: Following your suggestions, we have changed the title as follows: “Changes in body composition and body image perception in adolescent soccer players examined with repeated measurements during pre-season and in-season training.”

R2: The summary summarizes the objectives, methodological approach and main results; in my view it should include a reference highlighting adequately the practical implications of the results obtained.

A: Thank you for the suggestion: we have integrated the abstract with practical implications (lines 27-30).

R2: The introduction is well structured and contextualizes the relevance of the study; presenting a consistent theoretical background on body composition, body image and adolescence. With regard to the sources used, it is worth noting the inclusion of recent references, such as Gualdi-Russo et al. (2022) and Cabral et al. (2024); although some older references should be updated or supplemented (for example, Hausenblas & Downs, 2001) with recent meta-analyses or systematic reviews. In addition, it would be useful to incorporate studies that specifically address the relationship between body image and sporting ideal in football by reference to other collective and individual sports.

A: The reference reported has been replaced with a more recent one (Burgon et al., 2023), as you suggested. Furthermore, based on this reference, we have discussed the perceived and ideal body image in different sports groups (lines 296-299).

R2: As far as the method is concerned, the longitudinal design seems to be appropriate for the objectives of the study. The sample, although limited is acceptable (n=28), and a justified sampling with power analysis is used (GPower*). The tools used (MFBS and MBS) are validated and suitable for evaluating body image in adolescents. However, it would be useful to add information on the training of instrument implementers, the order in which scales are presented and the strategy used to minimize certain biases such as social desirability. It would also be relevant to clarify whether the scales are validated for the Italian population or a similar socio-cultural context and if intercoding reliability tests or internal consistency analysis were applied.

A: The information on the training of the operator and the order of presentation of two scales to participants are now added in the manuscript (lines 119-120 and 139). We would like to point out that in the two scales, the nine male body contour images are ordered from the thinnest (emaciated males) to the most adipose (obese males) in the MBS scale and from the thinnest (emaciated males) to the strongest (largest muscular figure) in the MFBS scale. In the two scales, each adjacent body contour of the same height increases the width (fat in MBS or muscle in MFBS) by 10%.

The scales proposed by Ralph-Nearman et al. (2018) have not been validated for Italians, but they were developed and validated on a sample of native-English speaking, in a similar socio-cultural context.

R2: The results section is well organized and the data are presented clearly through tables and repeated measures ANOVA analysis, which serves adequately for the purposes of the article. Interesting results can be seen in the evolution of body composition and image perception; however, there is a lack of graphs to facilitate comparative reading (for example, bar graphs or scatters between % body fat and body image discrepancy). Therefore, although significant differences are observed, the narrative could be strengthened by directly connecting physical changes with perceived changes in self-image, thus facilitating understanding of the type and degree of relationship between these variables.

A: Thanks for your positive comments and for your suggestions. We preferred to report data in tables rather than in graphs because tables are more comprehensive and allow the researchers to make comparisons and meta-analyses.

R2: The discussion is well articulated, integrating the results with existing literature and recognizing the complexity of the body ideal, also in the sports context. The analysis of the difference between social and sporting ideals is particularly appreciated. Precisely because of the interest of this type of association, it is recommended to enrich this section with recent studies that address the specific aesthetic pressures in adolescent male athletes and the psychological implications of body configuration in general and, more specifically, the role of ideal musculature between sportsmen and non-sportsmen on the one hand, the evolution of these relationships and, advancing hypotheses that could allow to tackle studies where the role of possible gender differences is relevant. It would also be desirable, in my opinion, to include a critical reflection on methodological limitations such as the small sample size, the lack of a control group or the absence of complementary qualitative analyses while more explicitly supplementing the possible pedagogical implications of the study.

A: Thank you for your comment and suggestions, which we have followed by integrating the text at lines 309-334, 343-348. While at lines 358-362, we have deepened the discussion regarding the methodological limitations.

R2: The conclusion adequately summarizes the main findings, highlighting the importance of sports practice in the physical and emotional well-being of adolescents in a socio-cultural context where lack of movement has already become a pandemic. However, it is too general and does not perhaps provide any new interpretative elements. It is suggested to expand it with concrete practical recommendations for trainers, educators and families, as well as proposing future lines of research, as the above-mentioned comparative analyses between genders and differences between sports disciplines or levels of competence. There could also be further reflection on the applicability of this methodology in educational or school contexts.

A: Following your guidance, we have strengthened the concepts with practical recommendations in lines 363-378.

R2: The bibliography is extensive, up-to-date and diverse, with a renewed contribution from the Introduction section. Of the total references used, more than 30% correspond to the last five years, which is in line with current editorial standards.

Based on the set of assessments made, I consider that the manuscript presents a relevant contribution to the field of physical education, health and sport psychology and it is recommended to accept the article with minor revisions.